# New-Onset Diabetes after Kidney Transplantation

**DOI:** 10.3390/medicina57030250

**Published:** 2021-03-08

**Authors:** Claudio Ponticelli, Evaldo Favi, Mariano Ferraresso

**Affiliations:** 1Nephrology, Dialysis and Transplantation, Fondazione IRCCS Ca’ Granda Ospedale Maggiore Policlinico, 20122 Milan, Italy; ponticelli.claudio@gmail.com; 2Renal Transplantation, Fondazione IRCCS Ca’ Granda Ospedale Maggiore Policlinico, 20122 Milan, Italy; mariano.ferraresso@unimi.it; 3Department of Clinical Sciences and Community Health, Università Degli Studi di Milano, 20122 Milan, Italy

**Keywords:** new-onset diabetes after transplantation, NODAT, diabetes, kidney transplantation, renal allograft, cardiovascular disease, immunosuppression, calcineurin inhibitor, mTOR inhibitor, steroid

## Abstract

New-onset diabetes mellitus after transplantation (NODAT) is a frequent complication in kidney allograft recipients. It may be caused by modifiable and non-modifiable factors. The non-modifiable factors are the same that may lead to the development of type 2 diabetes in the general population, whilst the modifiable factors include peri-operative stress, hepatitis C or cytomegalovirus infection, vitamin D deficiency, hypomagnesemia, and immunosuppressive medications such as glucocorticoids, calcineurin inhibitors (tacrolimus more than cyclosporine), and mTOR inhibitors. The most worrying complication of NODAT are major adverse cardiovascular events which represent a leading cause of morbidity and mortality in transplanted patients. However, NODAT may also result in progressive diabetic kidney disease and is frequently associated with microvascular complications, eventually determining blindness or amputation. Preventive measures for NODAT include a careful assessment of glucose tolerance before transplantation, loss of over-weight, lifestyle modification, reduced caloric intake, and physical exercise. Concomitant measures include aggressive control of systemic blood pressure and lipids levels to reduce the risk of cardiovascular events. Hypomagnesemia and low levels of vitamin D should be corrected. Immunosuppressive strategies limiting the use of diabetogenic drugs are encouraged. Many hypoglycemic drugs are available and may be used in combination with metformin in difficult cases. In patients requiring insulin treatment, the dose and type of insulin should be decided on an individual basis as insulin requirements depend on the patient’s diet, amount of exercise, and renal function.

## 1. Introduction

Kidney transplantation (KT) is the treatment of choice of end-stage renal disease. Over the years, significant advances in surgical techniques and medical care, as well as the development of new and more powerful immunosuppressive agents, have led to a progressive reduction of transplant-related mortality and morbidity [1]. However, compared to the general population, KT recipients have an increased risk of cardiovascular disease (CVD), infections, and malignancy [2,3]. Among the many complications that may occur during the post-transplant course, new-onset diabetes mellitus after transplantation (NODAT) deserves special consideration as it is commonly associated to metabolic disorders or infections and may result in diabetic kidney disease with nephrotic syndrome, impaired renal function or even premature allograft loss. In combination with pre- or co-existing conditions [4,5,6,7], NODAT may also contribute to the development and aggravation of major adverse cardiovascular events (MACE), which represent a leading cause of morbidity and mortality in this specific group of patients [8,9,10].

## 2. Definition of NODAT

The term NODAT was first used in 2003 and should be primarily and cautiously referred only to patients with no pre-transplant diagnosis of diabetes mellitus, on a stable maintenance immunosuppressive regimen, and no ongoing acute infections [11]. For epidemiological and clinical purposes, it is crucial to differentiate NODAT from other forms of post-transplant hyperglycemia, such as stress-induced hyperglycemia or transient post-transplant hyperglycemia. In this regards, the International Congress Guidelines [11], clearly state that a diagnosis of NODAT can be made in case of (a) fasting glucose ≥ 126 mg/dL (7 mmol/L) in more than one occasion; (b) random glucose ≥ 200 mg/dL (11.1 mmol/L) with symptoms; (c) two-hour glucose after a 75-g oral glucose tolerance test (OGTT) ≥ 200 mg/dL (11.1 mmol/L); or (d) hemoglobin A1C (HbA1c) ≥ 6.5%. Furthermore, according to the American Diabetes Association (ADA), if a patient has discordant results from two different tests, then the test result that is above the diagnostic cut-off point should be repeated, with consideration of the possibility of HbA1C assay interference. The diagnosis is made on the basis of the confirmed test. For example, if a patient meets the diabetes criterion of the HbA1C but not the one related to fasting plasma glucose, that person should nevertheless be considered to have diabetes [12].

Even though the criteria for NODAT proposed by the International Congress Guidelines and by the ADA are basically the same as for the general population, this specific form of type 2 diabetes is due to a progressive loss of adequate β-cell insulin secretion frequently on the background of insulin resistance. At least initially and often throughout their lifetime, these individuals do not need insulin treatment to survive. NODAT usually develops within three months after transplantation and has been reported to occur in 4% to 27% of KT recipients [13,14,15,16]. This incidence-related variability may be accounted for by age, ethnicity, lifestyle, length of follow-up, and presence of modifiable and non-modifiable risk factors. It is worth noticing that, also the definition of diabetes has evolved over the years. As such, probably the most important variable is the year of study initiation. Indeed, there has been a progressive decline in the incidence of NODAT since the 1990s.

## 3. Risk Factors for NODAT

In KT recipients, the risk factors for NODAT can be divided into modifiable and non-modifiable. The modifiable ones are basically the same as the general population for type 2 diabetes, i.e., overweight, dyslipidemia, arterial hypertension, poor physical exercise, and gestational diabetes. Post-transplant modifiable risk factors include the peri-operative stress caused by surgery and anesthesia [17], infections such as hepatitis C [18] or cytomegalovirus [19,20], vitamin D deficiency [21,22], hypomagnesemia [23], and immunosuppressive medications. Non-modifiable risk factors are a familial predisposition, older age, and ethnicity.

Glucocorticoid-associated hyperglycemia often occurs in conjunction with obesity and is usually due to acquired insulin resistance [24]. Multiple mechanisms are probably involved in the genesis of glucocorticoid-induced insulin resistance. Glucocorticoids exert their impact on metabolism through several different tissues in the body. In the presence of glucocorticoids, there is an increase in adiposity, as well as an increase in lipolysis, leading to elevated levels of free fatty acids in the circulation and an increase in insulin resistance [25]. In catabolic conditions, glucocorticoids inhibit protein synthesis and stimulate protein degradation in skeletal muscle by ubiquitin-proteasome-dependent proteolysis [26]. Muscle proteolysis releases both branched-chain and aromatic amino acids, which are associated with insulin resistance [27]. Glucocorticoids may induce post-receptor insulin signaling defects, also contributing to insulin resistance [28]. In the liver, glucocorticoids increase steatosis, thus causing insulin resistance, which is compounded by increased gluconeogenesis and hyperglycemia [29]. The bone is the site of osteocalcin production, driven by the insulin receptor. Osteocalcin normally participates in bone turnover, as well as it suppresses increases in adiposity and steatosis. These actions are inhibited by glucocorticoids [30]. Suppressed insulin secretion and β-cells apoptosis may concur with insulin resistance to glucocorticoid-associated hyperglycemia [31,32]. Diabetes develops after some weeks or months of oral glucocorticoids therapy. However, immediately after the administration of intravenous high-dose methylprednisolone, severe hyperglycemia can occur. It usually reverses spontaneously, but it may also herald the development of a true type 2 diabetes mellitus [33]. Attempts to reduce the toxicity of glucocorticoids in KT recipients included steroid-free regimens, usually based on the association of one calcineurin inhibitor (CNI) with a purine synthesis inhibitor, steroid-sparing regimens, based on low-dose prednisone (5–10 mg per day), or steroid-withdrawal after the first months after transplant.

Both the CNIs cyclosporine and tacrolimus have diabetogenic properties, which can be worsened by the concomitant use of high-dose glucocorticoids. CNIs can induce glucose intolerance by different mechanisms, including a decrease in insulin secretion [34], an increase in insulin resistance [35], and toxicity on β-cells [36]. The effects of tacrolimus are more profound and intense than cyclosporine [37]. It should be noted that the tacrolimus-specific binding protein (FKBP-12) is located in β-cells. Thus, tacrolimus can potentiate glucolipotoxicity in β-cells, possibly by sharing common pathways of β-cell dysfunction [38]. In contrast, the binding protein for cyclosporine (cyclophilin) is preferentially located in the heart, liver, and kidney.

Sirolimus and everolimus, the mammalian target of rapamycin (mTOR) inhibitors used in KT, have also been associated with glucose intolerance, especially when administered with tacrolimus [39]. The pathophysiology of mTOR inhibitor-induced hyperglycemia has not been totally clarified. Several mechanisms have been proposed. The deleterious effects of mTOR inhibitors on glucose metabolism might be induced by an increased insulin resistance secondary to a reduction of the insulin signaling pathway within the cells and/or by a reduction of insulin secretion via a direct effect on pancreatic β-cells [40,41], but probably the main mechanism rests on the induction of gluconeogenic pathway in the liver, which potentiates glucose intolerance [42].

## 4. Consequences of NODAT

CVD undoubtedly represents the most relevant and dreadful complication of NODAT [43,44,45,46,47,48]. However, especially in the long term, NODAT may also cause progressive diabetic kidney disease, including glomerular, tubulointerstitial, and/or vascular lesions, as well as proteinuria [49]. The pathological findings of NODAT are similar to those of typical diabetic kidney disease in native kidneys [50]. The earliest detectable light-microscopy change is a widening of the mesangium, due to increased accumulation of matrix. In more advanced phases, mesangial sclerosis and nodular accumulation of mesangial matrix develop. However, in comparison with diabetic nephropathy in native kidneys, NODAT is more frequently associated with vascular or tubulointerstitial changes caused by rejection, viral infection, drug nephrotoxicity, or poor quality of the donated kidney. The synergistic effect of these changes may explain a deleterious impact of NODAT on long-term graft survival [46,47,48,51].

NODAT may also be associated with other major complications affecting both small and large blood vessels, such as blindness and amputations. Advanced glycation end products (AGEs) excess is one of the most important mechanisms involved in the pathophysiology of these microvascular complications [52,53]. There have been a large number of new therapies tested in clinical trials for diabetic complications with, in general, rather disappointing results. Indeed, it remains to be fully defined as to which pathways in diabetic complications are essentially protective rather than pathological in terms of their effects on the underlying disease process [54]. Similar to the general population, diabetic complications are also commonly encountered in patients with NODAT, including ketoacidosis, neurologic, and ophthalmic complications, as well as recurrent hypoglycemia and shock [55].

## 5. Prevention of NODAT

To prevent the development of NODAT, some strategies should be followed. All KT candidates should undergo a thorough assessment of the risk of diabetes before enlistment. Patients at higher risk are those with CVD, hypertension, obesity, other metabolic syndrome components like dyslipidemia, polycystic ovary syndrome, a history of gestational or familiar diabetes. They should be checked by glycate hemoglobin levels determination and OGTT before transplantation.

Preventive measures include loss of overweight, lifestyle modification, reduced caloric intake, and physical exercise. In particular, careful energy income management and physical activity represent the cornerstone of every prevention strategy. Concomitant measures include aggressive control of lipids levels and blood pressure to reduce the risk of CVD. Hypomagnesemia is an independent risk factor of NODAT. Therefore, interventions targeting serum magnesium to reduce the risk of NODAT may be convenient [56]. In KT recipients, vitamin D may regulate the immune response and exert a number of protective effects from CVD, cancer, and infections [57]. In addition, it has been reported that low levels of 25(OH) vitamin D (≤10 ng/mL) at the time of transplantation were an independent risk factor for NODAT within the first post-transplantation year [58]. These data suggest that vitamin D levels should be monitored, and hypovitaminosis D should be corrected in KT recipients. Administration of insulin for the days of methylprednisolone treatment may be useful to prevent the possible development of overt diabetes [59,60]. Treatment with glinides such as mitiglinide (30 mg/day) or even better repaglinide (1.5 mg/day) during methylprednisolone therapy may also attenuate the risk of hyperglycemia [61]. 

Of much importance are the immunosuppressive strategies. For patients in high-risk groups, including certain ethnic backgrounds, older adults, the very young, and recipients with hepatitis C, consideration should be given to initiating immunosuppressive therapy with agents that are less diabetogenic [62,63]. Strategies for modifying immunosuppression include dose reduction or discontinuation of glucocorticoids, low-dose CNI associated with mycophenolate or azathioprine, withdrawal of CNI and their replacement with an mTOR inhibitor or belatacept. Tailoring immunosuppression regimens based on hepatitis C or cytomegalovirus serology may modify the risk of developing NODAT [64], as well as it does a steroid-free immunosuppressive protocol [65]. In patients with glucose intolerance, cyclosporine may reduce the risk of NODAT in comparison with tacrolimus, although also cyclosporine is a diabetogenic drug [66]. Instead, belatacept, a co-stimulation blocker acting on the CD28-CD80/86 pathway, does not induce hyperglycemia or NODAT [67]. In patients who developed NODAT under tacrolimus, an improvement of glucose intolerance and insulin discontinuation was obtained after conversion to belatacept [68]. The antiproliferative compounds mycophenolate and azathioprine are not diabetogenic, but these drugs have to be used together with glucocorticoids or CNIs. The potential diabetogenic effects of the most frequently used immunosuppressive medications are summarized in Table 1.

## 6. Treatment of NODAT

Management of diabetes mellitus in transplant patients is more challenging than the general population because of a greater risk of fluctuating kidney function. When selecting an appropriate oral hypoglycemic agent for recipients with impaired renal function, it is important to consider the possible danger of serious adverse effects.

Severe hypoglycemia may occur with sulfonylureas and glinides, as a result of overdose or drug-drug interaction with azole antifungals or other inhibitors of cytochrome P2C9, the main responsible of sulfonylureas metabolism [69]. Metformin may offer some advantage over other glucose-lowering agents, particularly with respect to the risk of hypoglycemia and weight gain [70]. In addition, metformin can exert beneficial effects on the kidney [71,72]. There has been concern about the possible development of life-threatening lactic acidosis when using metformin. However, many KT recipients used metformin without experiencing side effects [73], and two comprehensive reviews found no evidence of an increased risk of lactic acidosis with metformin compared to other anti-hyperglycemic treatments [74,75]. Restricting metformin administration to KTx patients with an estimated glomerular filtration rate > 45 mL/min/1.73 m^2^ is widely considered a safe option. Thiazolidinediones such as pioglitazone have negligible renal clearance. They can be used not only for preventing but also for treating NODAT effectively [76]. Potential side effects include oedema, congestive heart failure, and bone fractures [77]. Glucagon-like peptide-1 (GLP-1) inhibitors may reduce cardiovascular events and all-cause mortality, as well as the progression of renal disease in type 2 diabetes [78,79,80]. Even though, these drugs could induce some gastrointestinal side effects such as nausea, vomiting, and diarrhea, they may be very useful in overweight stabilized patients. A possible association with pancreatitis and/or malignancy has been a matter of concern. However, the evidence in favor of the hypothesis that GLP-1 inhibitors cause specific types of malignant disease or increase the risk for cancer is not convincing enough [81]. Dipeptidyl peptidase-4 inhibitors also called gliptins are generally considered to have a low rate of side effects. Concern about an increased risk of cardiovascular events or pancreatic cancer has not been confirmed by recent studies [82]. Sodium glucose cotransporter-2 (SGLT2) inhibitors such as canagliflozin, dapagliflozin, empagliflozin, ertugliflozin, and sotagliflozin inhibit glucose reabsorption in renal proximal tubular cells and facilitate its excretion in urine. As glucose is excreted, its plasma levels fall, leading to an improvement in all glycemic parameters. SGLT2 inhibition has been shown to reduce cardiovascular mortality and preserve kidney function in patients with type 2 diabetes [83,84].

However, potential concerns of SGLT2 inhibition in KT recipients include volume depletion and urinary tract infection (UTIs). In transplant recipients with pre-existing diabetes, stable allograft function, and no history of recurrent UTIs, empagliflozin was well tolerated, but the study was small and the follow-up too short to assess the potential prevention of cardiovascular events [85]. In a pilot study on 24 stable KT patients with diabetes, canagliflozin allowed to obtain a reduction of body weight, blood pressure, HbA1c levels, and requirement of other hypoglycemic agents without any episode of hypoglycemia or serious adverse events [86]. About 50% of patients with NODAT require insulin treatment. The dose and the type of insulin should be decided on an individual basis as insulin requirements depend on diet, amount of exercise, and renal function. Tapering of glucocorticoids also requires an adjustment of insulin dosage. Considering the fact that most subjects with NODAT actually have insulin resistance rather than insulin deficiency, the risk of severe hypoglycemia is relatively low. In this context, the importance of tight metabolic control cannot be emphasised enough. The target levels for glucose controls largely vary among diabetologists. It is likely is safer to obtain acceptable glucose levels rather than insist on “ideal” lower glucose levels. 

## 7. Conclusions

NODAT represents a relatively frequent and potentially severe complication of KT. Pathogenesis is multifactorial but chronic exposure to immunosuppression is actually considered one of the most important modifiable risk factors. Further studies are needed to better understand the impact of currently available prevention and management strategies.

## Figures and Tables

**Table 1 medicina-57-00250-t001:** Main immunosuppressive medications and their diabetogenic potential.

Drug	Diabetogenic Effect
Maintenance immunosuppression:	
Glucocorticoids	+++++
Tacrolimus	++++
Cyclosporine	+++
Sirolimus	+++
Everolimus	++
Azathioprine	-
Mycophenolic acid	-
Induction immunosuppression:	
Basiliximab	-
Rabbit anti-thymocyte globulin	-
Rituximab	-

## Data Availability

Not applicable.

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
