# Peer review of "New-Onset Diabetes after Kidney Transplantation"

_medicina, 2021, doi:10.3390/medicina57030250_

Round 1
Reviewer 1 Report
The authors submit a review focused on new onset of diabetes after kidney transplantation (NODAT). The manuscript comprises different aspects of NODAT including definition, prevalence, risk factors, complications, prevention and treatment. Therefore, though it spans 4 and a half of print text pages (without references), the analysis does not go too deep into the pathophysiology and does not contain any figures, schemes or tables for better understanding. The paper is probably designed for those who are more or less new in the area. Most information is up to date, there are 83 references. The review is systematic and scientifically correct; however, it is a bit monotonous and lacks any scheme, e.g. regarding principal pathophysiologic factors or risk factors or any tables, e.g. regarding treatment options.
Minor comments:
Title: the paper is focused only on NODAT after kidney transplantation, therefore, this should appear also in the title.
The 1st and 2nd sentences in the introduction are difficult to read. They probably need rephrasing.
There is a definition of NODAT in section 2. However, other important terms in the context and frequently discussed in the literature should be explained, namely transient posttransplant hyperglycemia and post-transplant diabetes. The term NODAT can be used primarily if subjects in whom the state of glucose tolerance had been appropriately tested and a diabetes had been excluded prior to transplantation.
It is important to say that according to ADA and International Congress Guidelines, the criteria for diabetes diagnosis are the same as for the whole population.
It would be better to classify the risk factors as non-modifiable and modifiable, better then as traditional and non-traditional.
The most important modifiable factor is glucocorticoid therapy. The authors should pay more attention to this topic. They should also review the outcomes in corticoid-free, corticoid sparing and late corticoid withdrawal regimens. Also, it should be more in detail addressed insulin resistance (and rather high insulin levels) rather than insulin deficiency.
There should be also some recommendation regarding the diabetogenic effects of CNIs and their blood levels.
Section 4: the 1st sentence requires rephrasing.
Prevention of NODAT: it should be stressed, that the most important factors are careful energy income management and physical activity. For patients with high NODAT risk, specific immunosuppressive regimens should be considered, especially glucocorticoid-free or sparing. Rosiglitazone has been withdrawn from the EU market and is very rarely used in the USA.
Therapy: Here, a table for better orientation would be helpful. Again – rosiglitazone has been used only rarely at present time. GLP-1 analogues: do they really have such serious gastrointestinal effects and risk of cancer? Their use may be very useful in overweight stabilized patients.
The risk of hypoglycemia: Most subjects with NODAT are insulin resistant and the risk of hypoglycemia is not so high. Tight metabolic control is very important.
The last sentence is absolute nonsense. Early morning blood glucose level of 65 mg/dL (3,6 mmol/l) should be avoided in any subject.
Author Response
Please, see the attachment.

Reviewer 2 Report
This is an excellent review and state-of-the-art on a very important topic, NODAT.
I have no MAJOR CONCERNS.
MINOR COMMENTS and suggestions:
They are added in yellow notes on the original pdf.
- If you maintain the title of the paper as "NODAT in transplantation" and not "... in renal transplantation", I suggest to the authors to add some article/reference on NODAT in organ transplantation, i.e., HIGLU Study, Martínez-Castelao A et al, Transplant Proc 2005, with some data with regards to NODAT in other non-renal organ transplantation.
- - In my modest opinion a brief paragraph of conclusions may be added at the end of the discussion.

Author Response
Please, see the attachment.

Round 2
Reviewer 1 Report
The manuscript has much improved.